# Peripheral Beta-2 Adrenergic Receptors Mediate the Sympathetic Efferent Activation from Central Nervous System to Splenocytes in a Mouse Model of Fibromyalgia

**DOI:** 10.3390/ijms24043465

**Published:** 2023-02-09

**Authors:** Shiori Yamashita, Naoki Dozono, Shota Tobori, Kazuki Nagayasu, Shuji Kaneko, Hisashi Shirakawa, Hiroshi Ueda

**Affiliations:** 1Department of Molecular Pharmacology, Graduate School of Pharmaceutical Sciences, Kyoto University, 46-29 Yoshida-Shimoadachi-cho, Sakyo-ku, Kyoto 606-8501, Japan; 2Department of Pharmacology and Therapeutic Innovation, Nagasaki University Institute of Biomedical Sciences, Nagasaki 852-8521, Japan; 3Graduate Institute of Pharmacology, National Defense Medical Center, Nei-hu, Taipei 114201, Taiwan

**Keywords:** fibromyalgia, adrenergic receptors, central nervous system, AcGP model, splenocytes, sympathetic efferent, pain development

## Abstract

Abnormalities in the peripheral immune system are involved in the pathophysiology of fibromyalgia, although their contribution to the painful symptoms remains unknown. Our previous study reported the ability of splenocytes to develop pain-like behavior and an association between the central nervous system (CNS) and splenocytes. Since the spleen is directly innervated by sympathetic nerves, this study aimed to examine whether adrenergic receptors are necessary for pain development or maintenance using an acid saline-induced generalized pain (AcGP) model (an experimental model of fibromyalgia) and whether the activation of these receptors is also essential for pain reproduction by the adoptive transfer of AcGP splenocytes. The administration of selective β2-blockers, including one with only peripheral action, prevented the development but did not reverse the maintenance of pain-like behavior in acid saline-treated C57BL/6J mice. Neither a selective α1-blocker nor an anticholinergic drug affects the development of pain-like behavior. Furthermore, β2-blockade in donor AcGP mice eliminated pain reproduction in recipient mice injected with AcGP splenocytes. These results suggest that peripheral β2-adrenergic receptors play an important role in the efferent pathway from the CNS to splenocytes in pain development.

## 1. Introduction

Fibromyalgia (FM) is a common syndrome characterized by chronic widespread pain accompanied by various symptoms, such as fatigue, unrefreshed sleep, or cognitive dysfunction [1,2]. Limited knowledge was available on the detailed pathogenic mechanisms of FM, making it difficult to identify plausible biomarkers and therapeutic targets for FM-specific treatments. It is hypothesized that the pain from FM is caused by an enhancement of the neuronal response in the nociceptive pathways of the central nervous system (CNS) [2]. This phenomenon is known as central sensitization [3]. For example, functional magnetic resonance imaging revealed that the patients with FM had greater activity in the pain-processing areas of their brain in response to painful or non-painful stimuli than the controls [4]. Moreover, glial cells are known to contribute to CNS inflammation in central sensitization-related chronic pain [5], which is in agreement with the activation of glial cells in the brains of patients with FM [6]. On the other hand, spontaneous activity in silent C nociceptors [7] and increased sensory nerve excitability [8] have been also reported in the peripheral sensory pathways of FM patients.

In addition, some reports have suggested that dysregulation of the peripheral immune system may be involved in FM pathophysiology. FM is more prevalent in patients with autoimmune diseases, such as systemic lupus erythematosus, rheumatoid arthritis, and autoimmune thyroid disease [9]. At the molecular level, blood concentrations of some cytokines and chemokines increased and positively correlated with symptom severity in patients with FM [10,11]. Recent findings indicated that differentially expressed genes, miRNAs and long non-coding RNAs associated with dysregulated immune system were found between FM patients and healthy control, which may play an important role in the pathogenesis of FM [12,13]. However, it is unclear whether abnormal states of the immune system lead to an increase in neuronal excitability of nociceptive circuits and cause painful symptoms in FM or whether other pathogenic factors affect the patient’s immunity.

To answer these questions, we need to use FM-like generalized pain disease models. So far, there are several FM-like animal models showing generalized pain, which is caused by dual muscular injections of acid saline [14], vagotomy [15], sound stress [16], intermittent cold stress [17], intermittent psychological stress (empathy) [18] and reserpine [19]. The acid-induced generalized pain model that we called an AcGP model [20] was found to share some pathophysiological and pharmacotherapeutic features with FM patients in clinic in terms of generalized or bilateral hyperalgesia by dual acid saline injections into the gastrocnemius muscle on the one side [14], female-predominant pain [20], and pregabalin-reversible pain [21]. In our previous study, bilateral hyperalgesia was elicited over a long period of time, despite the absence of obvious histological changes in the gastrocnemius muscle. In addition, local lidocaine administration or ipsilateral dorsal rhizotomy did not reverse contralateral hyperalgesia [14]. Thus, AcGP seems to depend on the sensitization of nociceptive pathways itself rather than peripheral inflammation or continuous input from primary afferent nociceptors at the acid injection site. Indeed, we obtained evidence that the AcGP model shows an association between central sensitization and splenic T-cells in mice [20]. The systemic injection of splenocytes or CD4-positive T-cells derived from AcGP mice causes a significant hyperalgesia in naïve mice, and the intracerebroventricular administration of minocycline (a tetracycline antibiotic and well-known microglia inhibitor) [22] in AcGP donor mice diminished pain reproduction in recipient mice transferred with AcGP splenocytes. These observations suggest that there is a relationship between the brain mechanisms and splenocyte activation in the AcGP model.

The sympathetic nervous system directly modulates the immune function in the spleen. The splenic nerve arises from the celiac ganglion and consists of sympathetic fibers that enter the hilus of the spleen along with the splenic arteries [23,24]. Noradrenaline released from the endings of sympathetic splenic fibers causes contraction of the spleen, vasoconstriction, and modulation of various immune cell responses to exogenous stimuli [25,26,27]. In patients with FM, an elevation of sympathetic activity with a low parasympathetic tone has been described in many clinical studies [28]. Therefore, it is worth confirming that the sympathetic nervous system acts as an efferent pathway through which the CNS communicates and cooperates with the spleen to augment nociception in an AcGP model. More specifically, it is known that the spleen expresses β-adrenergic receptors, and the β2 subtype is more abundant than β1 [29,30]. Sympathetic input through β2-adrenergic receptors can alter the proliferative activity and cytokine expression of splenic lymphocytes during the immune response. In this study, we aimed to investigate whether adrenergic receptors were involved in the development and maintenance of pain-like behavior in AcGP mice.

## 2. Results

### 2.1. β2-Adrenergic Receptors Are Involved in the Development but Not in the Maintenance of the Acid-Induced Mechanical Hypersensitivity

As previously reported, repeated acid saline (pH 4.0) injections into the gastrocnemius muscle significantly decreased the paw withdrawal threshold (PWT) to mechanical stimuli 5 days after the second acid injection (post day 5, P5) [20]. In contrast, control saline (pH 7.2) injections had no effect on PWT.

To determine whether the activation of β2-adrenergic receptors contributed to the development of acid-induced mechanical hypersensitivity, butoxamine, a selective β2-blocker, was administered intraperitoneally in mice, which was followed by a second acid injection after 30 min (Figure 1A). Pre-treatment with 10 and 30 mg/kg butoxamine prevented the significant decrease in PWT at P5 (AcGP + 10 mg/kg butoxamine (8.24 ± 0.513, *n* = 7), AcGP + 30 mg/kg butoxamine (8.71 ± 0.646, *n* = 9) vs. AcGP + Vehicle (5.42 ± 0.301, *n* = 18); *p* = 0.0006, *p* < 0.0001, respectively) (Figure 1B). We also confirmed that carazolol had antagonistic actions on β2-adrenergic receptors, although it has a lower permeability at the blood–brain barrier than butoxamine [31]. A similar dose-dependent preventive effect was observed on the decrease in PWT at P5 (AcGP + 1 mg/kg carazolol (7.93 ± 0.287, *n* = 8), AcGP + 10 mg/kg carazolol (8.25 ± 0.225, *n* = 8) vs. AcGP + Vehicle (6.39 ± 0.265, *n* = 12); *p* = 0.0111, *p* = 0.0017, respectively) (Figure 1C). These results indicate that peripheral β2-adrenergic receptors are involved in the development of acid-induced mechanical hypersensitivity.

Next, to investigate the effect of β2-adrenergic receptor blockade after the development of hypersensitivity, the drugs were administered from P5 to P11 (Figure 1D). At P12, the PWT of AcGP mice post-treated with butoxamine or carazolol was still low, as was the case with vehicle treatment, indicating that β2-adrenergic receptors have no role in the maintenance of acid-induced hypersensitivity (AcGP + 10 mg/kg butoxamine (6.83 ± 0.588, *n* = 6) vs. AcGP + Vehicle (6.03 ± 0.525, *n* = 5); AcGP + 10 mg/kg carazolol (5.89 ± 0.764, *n* = 6) vs. AcGP + Vehicle (5.65 ± 0.360, *n* = 6); *p* = 0.643, *p* = 0.987, respectively).

### 2.2. Neither α1-Adrenergic nor Muscarinic Receptors Contribute to the Acid-Induced Mechanical Hypersensitivity

It is known that a sympathetic nerve stimulation or an administration of catecholamines cause a contraction of splenic capsule and vasoconstriction, which are mediated by the activation of α1-adrenergic receptors. To investigate whether α1-adrenergic receptors also contribute to acid-induced mechanical hypersensitivity, we administered prazosin, an α1-blocker, 30 min before the second acid injection (Figure 2A). The results showed that prazosin did not prevent the development of acid-induced mechanical hypersensitivity (AcGP + 10 mg/kg prazosin (6.24 ± 0.359, *n* = 8) vs. AcGP + Vehicle (6.21 ± 0.354, *n* = 8); *p* = 0.999). In addition, acetylcholine (ACh) did not seem to have a major effect on pain-like behavior because butylscopolamine (an anticholinergic drug) did not affect the development of mechanical hypersensitivity in the AcGP model (AcGP + 15 mg/kg butylscopolamine (5.98 ± 0.300, *n* = 4) vs. AcGP + Vehicle (5.72 ± 0.447, *n* = 8); *p* = 0.914) (Figure 2B).

### 2.3. Adoptive Transfer of Splenocytes from AcGP Mice Stimulated by β2-Adrenergic Receptor Activation Reproduces the Mechanical Hypersensitivity in Naïve Recipient Mice

As mentioned above, we previously reported the reproduction of pain-like behavior in naïve mice by the intravenous administration of splenocytes derived from AcGP mice. To examine whether β2-adrenergic receptor activation in donor AcGP mice is necessary for splenocytes to induce mechanical hypersensitivity in recipient mice, butoxamine was administered to donor mice before the second acid injection, and splenocytes were prepared and transferred at P5 (Figure 3A). The results showed that the adoptive transfer of splenocytes from AcGP mice pre-treated with butoxamine did not decrease the PWT in recipient mice (AcGP + 30 mg/kg butoxamine (8.21 ± 0.358, *n* = 8) vs. AcGP + Vehicle (5.61 ± 0.282, *n* = 8); *p* < 0.0001) (Figure 3B). In contrast, pre-treatment with prazosin had no effect on the PWT decrease in recipient mice (AcGP + 10 mg/kg prazosin (6.32 ± 0.343, *n* = 8) vs. AcGP + Vehicle (6.07 ± 0.245, *n* = 8); *p* = 0.945) (Figure 3C). These results indicate that β2-adrenergic receptor activation stimulates splenocytes in AcGP mice and enables them to reproduce mechanical hypersensitivity in recipient mice.

## 3. Discussion

In the present study, we found that treatment with a selective β2-blocker butoxamine just before the second acid injection prevented the decrease in PWT at P5. The prevention of pain-like behavior also occurred by the peripheral β2-blocker carazolol, suggesting that the pain development stimulated by β2-adrenergic receptors occurred in peripheral tissue in AcGP mice not in the sensory pathways of the CNS. On the contrary, β2-adrenergic receptors did not seem to be involved in pain maintenance in AcGP mice because the drug treatments did not reverse the decrease in PWT when they were administered from P5 to P11. We also showed that α1-adrenergic or muscarinic receptor blockade at the time of the second acid injection did not affect the decrease in the PWT. In addition, pain reproduction in recipient mice induced by the adoptive transfer of AcGP mice-derived splenocytes did not occur when pain development in donor mice was inhibited by β2-adrenergic receptor blockade. This observation suggests that the contribution of β2-adrenergic receptor activation to pain development in AcGP mice is mediated by activated splenocytes, although it remains elusive whether these results could be reproduced in other FM models or other animal species.

According to quantitative autoradiography, β2-adrenergic receptors are expressed more as the β1 subtype in the spleen [29]. In addition, lymphocytes predominantly express the β2 subtype of β-adrenergic receptors [30]. Our previous study showed that the adoptive transfer of CD4+ T lymphocytes separated from AcGP mouse splenocytes caused pain reproduction in recipient mice [20]. Furthermore, this study showed that when donor AcGP mice were treated with butoxamine, their splenocytes could not induce pain even when transplanted into the recipient mice. Therefore, β2-adrenergic receptor-expressing CD4+ T lymphocytes may be essential for acid-induced pain development. However, it is possible that β2 receptors expressed at a location other than splenocytes are involved in pain induction. It has been reported that adrenaline can sensitize primary afferent nociceptors and cause hyperalgesia via the activation of free nerve ending β receptors [32]; hence, the β-adrenergic receptor can also enhance signal transduction from the gastrocnemius muscle to the CNS induced by the second acid injection. Therefore, future experiments, such as immunohistochemistry for β2 receptors, intramuscular co-injection of β2 antagonists with acid saline, or splenic denervation, are needed to elucidate which organs or cells are stimulated via β2 receptors and lead to pain development in the AcGP model.

Several reports have indicated that the activation of β2 receptors on immune cells suppresses the immune response. For example, activation of the β2–Gαs–cAMP pathway in immune cells results in immunosuppression [33]. It has also been reported that electrical stimulation of sympathetic nerves results in the release of noradrenaline in the spleen, which activates β2 receptors on ChAT-positive T cells and promotes Ach release. Ach release in turn acts on α7nACh receptors on splenic macrophages and suppresses TNF-α release from macrophages, indicating that β2 receptor stimulation is immunosuppressive [34,35]. In contrast, the stimulation of β2 receptors in immune cells is also known to cause immunopotentiation. This effect may be due to Gs-independent pathways. In this context, downstream, the G protein-dependent activation of protein kinase A (PKA) causes desensitization by phosphorylating β2 receptors and leads to impairment of their binding affinity to Gs and enhancement of their binding affinity to Gi. Therefore, the effect of β2 receptor activation on adenylate cyclase was reversed. Moreover, high-agonist concentrations can activate mitogen-activated protein kinase (MAPK) pathways in a G protein-independent manner, which induces the nuclear translocation of transcription factors to change the gene expression profile [36]. Thus, the effects of β2 receptor activation are likely stimulus-dependent. Furthermore, it has been reported that sympathetic nerves alter the pattern of co-transmitter release depending on their firing pattern [37]. In the AcGP model, both co-transmitters and adrenaline from sympathetic nerves modulate the immune properties in a complex manner. Future investigations are needed regarding the phenotypic changes in splenocytes at both the transcriptional and translational levels that contribute to pain formation and whether these changes are directly attributable to β2 receptor activation.

Our previous findings showed that the mRNA expression of CD4 increased in the spleen of AcGP mice and that the transplantation of CD4-positive T cells from donor mice into naïve mice induced pain. It has been shown that β2 receptors are more abundant on naïve T cells and Th1 cells and less abundant on Th2 cells than those on other CD4-positive T cells [30]. Moreover, several reports have suggested that Th1 cells contribute to pain in patients with FM [38,39]. Thus, the main active immune system in FM is cellular immunity involving T cells rather than humoral immunity involving B cell-derived IgG antibodies. As the degree of Th17 cell activation downstream of β2 receptors has been reported to be altered in other disease models [40,41], the involvement of Th17 in the AcGP model used in this study should be examined in the future. It is also unclear how immune cells activated in the spleen affect each of the central and peripheral nociceptive pathways, both of which have been indicated to be sensitized in an AcGP model [42,43,44].

The present study suggests that the activation of the β2 receptor contributes to pain development. However, β2 agonists are known to ameliorate pain-like behavior in an animal model of neuropathic pain [45,46,47], which seems to contradict our finding. This inconsistency may be due to the difference in the process of chronic pain development. In the context of neuropathic pain, the adoptive transfer of splenocytes derived from chronic constriction injury (CCI) animals to naïve ones does not cause pain reproduction in recipients [48], indicating splenocytes are not involved in the pathology of neuropathic pain unlike in the AcGP model. The effect of β2 agonists in neuropathic pain may be mediated by adrenoceptors expressed at a location other than splenocytes. It is important to distinguish these chronic pain conditions and find therapeutic targets for each of them.

## 4. Materials and Methods

### 4.1. Animal Experiments

All experiments were conducted in accordance with the ethical guidelines of the Kyoto University Animal Research Committees (Approval number: 19-38) and Nagasaki University Animal Research Committees (Approval number: 1607201325-8). In addition, the present study has been performed in order to comply with ARRIVE guidelines [49]. Male C57BL/6J mice (6–12 weeks old) from Japan SLC (Shizuoka, Japan) were used, as reported previously [20,44,50]. They were housed at 22 ± 2 °C under a 12 h light/dark cycle with a standard laboratory diet and water available ad libitum. The total number of animals used in this study was 290. Group sizes for each experiment are described in Table 1. For the purpose of randomization, the paw withdrawal thresholds of each animal were measured before experiments, and animals were assigned so that the control and treatment groups had approximately equal mean paw withdrawal thresholds. For the blinding, investigators in charge of project and experiment designing and the conduct of experiments are independent to each other. Data analyses after the experiments were performed by all authors in the presence of non-authors.

### 4.2. AcGP Mouse Model

The AcGP model was developed according to a previous study [14]. Briefly, mice received a unilateral injection of 20 μL acid saline (pH 4.0) into the gastrocnemius muscle of the left hindlimb through a 27-gauge needle under isoflurane (4%) anesthesia. Injections were conducted twice, 5 days apart (Day 0/D0 and Day 5/D5). Mice in the control group were injected with pH 7.2 saline.

### 4.3. Drug Treatments

All drugs were dissolved in saline and injected intraperitoneally. Butoxamine hydrochloride (B1385; Sigma-Aldrich, St Louis, MO, USA), carazolol (C2578; Tokyo Chemical Industry, Tokyo, Japan), prazosin hydrochloride (162-14681; Wako, Osaka, Japan), and (−)-scopolamine N-butyl bromide (S7882; Sigma-Aldrich, St Louis, MO, USA) were used in this study.

### 4.4. Behavioral Test

Mice were placed in an individual chamber on a wire mesh grid floor and acclimatized for over 2 h before the trials began. Mechanical stimuli were applied to the plantar surface of the right hind paw using an electronic digital von Frey anesthesiometer and a rigid tip (model 2390, 90 g probe; IITC Inc., Woodland Hills, CA, USA). The pressure on the plantar surface was recorded until the mice showed flexor reflex, and the maximum value was determined as the mechanical paw withdrawal threshold (PWT).

### 4.5. Isolation and Adoptive Transfer of Splenocytes

Splenocytes were prepared according to a previous study [20]. At D5, AcGP mice were euthanized by cervical dislocation, and the spleen was removed immediately. The spleen was placed in 3 mL of ice-cold RPMI 1640 medium (Gibco, Grand Island, NY, USA) containing 2% fetal bovine serum (FBS) and minced using the plunger of a 1 mL injection syringe. The suspension was filtered through a 70 μm mesh cell strainer (Corning, Glendale, AZ, USA) into a 50 mL tube and washed with 5 mL of ice-cold PBS containing 2% FBS. The single-cell suspension was centrifuged at 500× *g* for 5 min at 4 °C. After removal of the supernatant, the pellet was incubated in 3 mL of red blood cell lysis buffer (Abcam, San Diego, CA, USA) for 3 min at room temperature, and hemolysis was stopped by adding 5 mL of ice-cold PBS containing 2% FBS. Splenocytes were pelleted by centrifugation at 500× *g* for 5 min at 4 °C, washed with ice-cold PBS containing 2% FBS, and resuspended in 3 mL of the same buffer. The suspension was then filtered through a cell strainer. A 200 μL aliquot of the suspension containing 1 × 10^7^ cells was injected intravenously into the tail vein of each naïve recipient mouse.

### 4.6. Statistical Analysis

All statistical analyses were performed using Prism 9 (GraphPad Software, San Diego, CA, USA). The Shapiro–Wilk test was used for testing the normality of the data. One-way analysis of variance (ANOVA) with Tukey’s post hoc test was used for experiments with drug pre-treatment, and two-way ANOVA with Tukey’s post hoc test was used for experiments with drug post-treatment or adoptive transfer of splenocytes. In all cases, a *p*-value < 0.05 was considered statistically significant. Data are presented as mean ± SEM.

## 5. Conclusions

In the present study, peripheral β2-adrenergic receptor blockade prevented both pain development and pain reproduction in AcGP model. Our work reveals a crucial role of peripheral β2-adrenergic receptors in efferent activation from CNS to splenocytes. These findings provide further insight into the mechanism of developing generalized pain in FM.

## Figures and Tables

**Figure 1 ijms-24-03465-f001:**
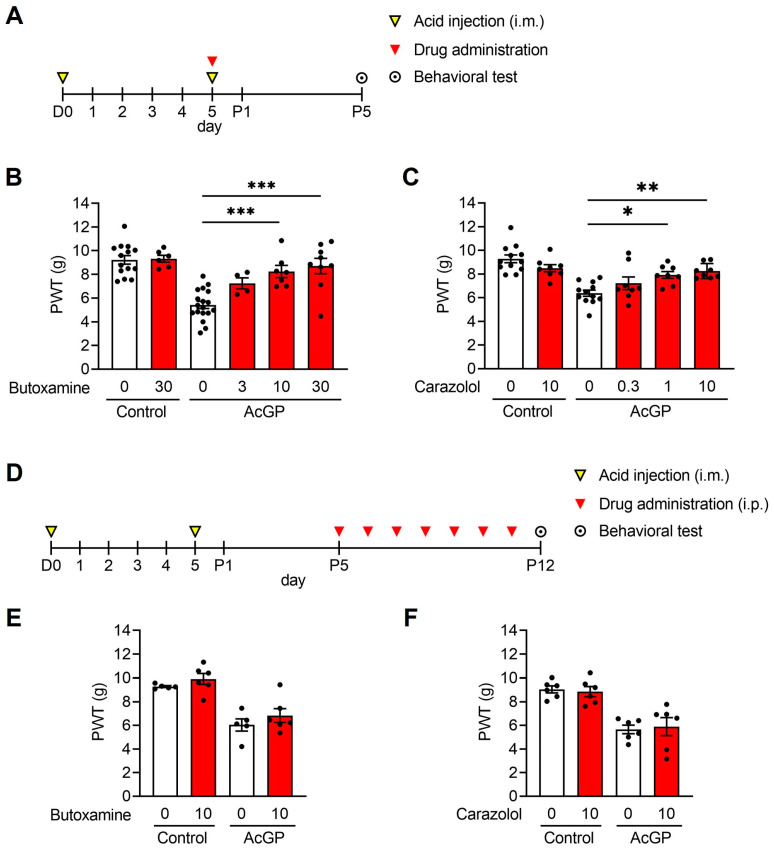
Effects of β2-adrenergic receptor blockade on the development and maintenance of pain-like behavior in AcGP mice. (**A**) Experimental time course for drug pre-treatment and the behavioral test; (**B**,**C**) Paw withdrawal threshold to mechanical stimuli of AcGP mice pre-treated with butoxamine (**B**) or carazolol (**C**); (**D**) Experimental time course for drug post-treatment and the behavioral test; (**E**,**F**) Paw withdrawal threshold to mechanical stimuli of AcGP mice post-treated with butoxamine (**E**) or carazolol (**F**). Data are presented as mean ± SEM. (**B**) *n* = 4–18; (**C**) *n* = 8–12; (**E**) *n* = 5–6; (**F**) *n* = 6. * *p* < 0.05, ** *p* < 0.01, *** *p* < 0.001 for one-way ANOVA with Tukey’s post hoc test (**B**,**C**). AcGP, acid saline-induced generalized pain; PWT, paw withdrawal threshold.

**Figure 2 ijms-24-03465-f002:**
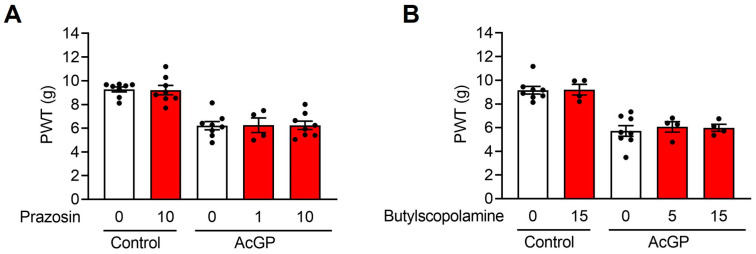
Neither α1-adrenergic nor muscarinic receptor blockades affect the development of pain-like behavior in AcGP mice. (**A**,**B**) Paw withdrawal threshold to mechanical stimuli of AcGP mice pre-treated with prazosin (**A**) or butylscopolamine (**B**). Data are presented as mean ± SEM. *n* = 4–8. AcGP, acid saline-induced generalized pain; PWT, paw withdrawal threshold.

**Figure 3 ijms-24-03465-f003:**
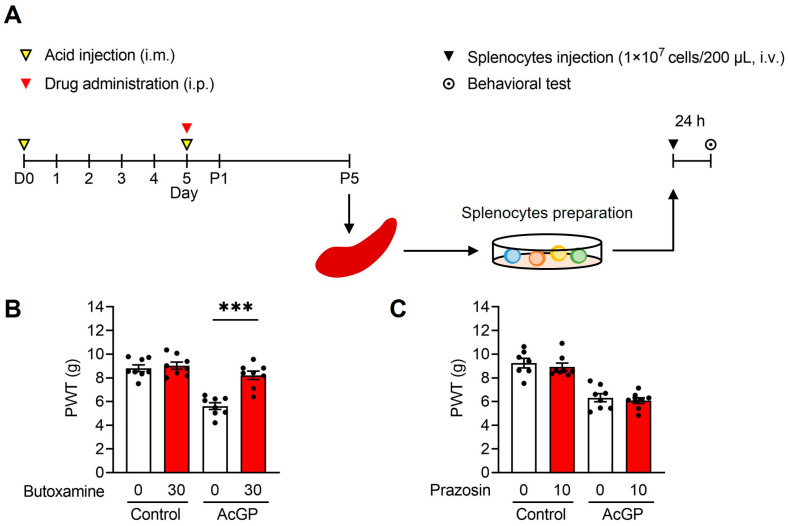
Effects of β2-adrenergic or α1-adrenergic receptor blockades on the pain reproduction in recipient mice transferred with AcGP splenocytes from donor AcGP mice. (**A**) Experimental time course for the adoptive transfer of splenocytes and the behavioral test; (**B**,**C**) Paw withdrawal threshold to mechanical stimuli of recipient mice transferred with splenocytes from AcGP mice pre-treated with butoxamine (**B**) or prazosin (**C**). Data are presented as mean ± SEM. *n* = 8 (**B**); *n* = 7–8 (**C**). *** *p* < 0.001 for two-way ANOVA with Tukey’s post hoc test. AcGP, Acid saline-induced generalized pain; PWT, paw withdrawal threshold.

**Table 1 ijms-24-03465-t001:** Number of animals used in each experimental group.

Figure	Group		Total Number of Animals Used	Sample Size	Number of Animals Excluded ^1^
Figure 1B		Butoxamine concentration (mg/kg)			
	Control	0	16	14	2
		30	6	6	0
	AcGP	0	18	18	0
		3	4	4	0
		10	8	7	1
		30	9	9	0
Figure 1C		Carazolol concentration (mg/kg)			
	Control	0	12	12	0
		10	8	8	0
	AcGP	0	12	12	0
		0.3	8	8	0
		1	8	8	0
		10	8	8	0
Figure 1E		Butoxamine concentration (mg/kg)			
	Control	0	5	5	0
		30	6	6	0
	AcGP	0	5	5	0
		30	6	6	0
Figure 1F		Carazolol concentration (mg/kg)			
	Control	0	6	6	0
		10	7	6	1
	AcGP	0	6	6	0
		10	6	6	0
Figure 2A		Prazosin concentration (mg/kg)			
	Control	0	8	8	0
		10	8	8	0
	AcGP	0	8	8	0
		1	4	4	0
		10	8	8	0
Figure 2B		Butylscopolamine concentration (mg/kg)			
	Control	0	8	8	0
		15	4	4	0
	AcGP	0	8	8	0
		5	4	4	0
		15	4	4	0
Figure 3B		Butoxamine concentration (mg/kg)			
	Control	0	8	8	0
		30	8	8	0
	AcGP	0	8	8	0
		30	9	8	1
Figure 3C		Prazosin concentration (mg/kg)			
	Control	0	7	7	0
		10	8	8	0
	AcGP	0	8	8	0
		10	8	8	0

^1^ Animals which died at the time of drug injection or anesthesia were excluded.

## Data Availability

All data are included in the manuscript.

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
