# Peer review of "Peripheral Beta-2 Adrenergic Receptors Mediate the Sympathetic Efferent Activation from Central Nervous System to Splenocytes in a Mouse Model of Fibromyalgia"

_ijms, 2023, doi:10.3390/ijms24043465_

Round 1

Reviewer 1 Report

Thank you for leaving a great achievement in pain science.

Discussion

The design and reporting are generally good, but lack limitations and implications to conclude.

Methods

1. References are required to confirm that this is a systematic experimental method.

2. Please write the product name, company, and country for all devices, drugs, and software.

Author Response

Replies to the comments from the reviewer

Thank you very much for your time and his/her encouraging comments and constructive suggestions with regard to our manuscript. Those comments are helpful for us to revise and improve our manuscript. We have studied comments carefully and tried our best to revise and improve the manuscript according to the Reviewers’ comments. Revised portion is highlighted in RED FONT in the revised manuscript. We hope that the corrections will meet with approval. The main corrections in our manuscript and the responses to the Reviewers’ comments are as follows:

Reviewer 1:

Discussion

The design and reporting are generally good, but lack limitations and implications to conclude.

Response;

We agree with the reviewers’ concern and apologize for incomplete discussion. We have added some limitations and implications to Discussion Section.

Methods

  1. References are required to confirm that this is a systematic experimental method.

Response;

As the experimental method performed in this study was determined based on similar previous researches in this field [16, 18]. We added the description to Method Section (page 8-9).

  1. Please write the product name, company, and country for all devices, drugs, and software.

Response;

According to the suggestion by the reviewer, we added the description about the country of all devices, drugs, and software to Method Section (page 8-10).

Reviewer 2 Report

In this study, the authors examined whether adrenergic receptors are necessary for pain development or maintenance in an experimental model of fibromyalgia and whether activation of these receptors is also essential for pain reproduction by adoptive transfer of AcGP splenocytes. They concluded that peripheral β2-adrenergic receptors play an important role in the efferent pathway from the CNS to splenocytes in pain development. These experiments are preliminary data. The major point is that the underlying mechanisms of the protective effect of peripheral β2-adrenergic receptors blockers remain elusive. Moreover, there are several concerns that should be addressed.

1. Reporting animal usage: Although this is generally well reported in this study, there are some extra details required in order to comply with ARRIVE guidelines (Percie du Sert, Hurst et al. 2020). To rectify this, the authors need to:

•     Abstract: State the species and strain of animals used in this study in the abstract

•     Introduction: Explain how and why the animal species and model being used can address the scientific objectives and the study’s relevance to human biology.

•     Methods: State the total number of animals used in the study, as well as clearly indicate group sizes for each experiment, and the rationale for using the number and strain of animals chosen (e.g. sample size calculations). State whether randomisation was used to allocate experimental units to control and treatment groups. If done, provide the method used to generate the randomisation sequence

•     Blinding: Describe who was aware of the group allocation at the different stages of the experiment (during the allocation, the conduct of the experiment, the outcome assessment, and the data analysis).

•     Results: For each experimental group, report any animals, experimental units or data points not included in the analysis and explain why. For each experiment conducted, including independent replications, report: a. Summary/descriptive statistics for each experimental group, with a measure of variability where applicable (e.g. mean and SD, or median and range). b. If applicable, the effect size with a confidence interval.

•     Discussion: Comment on whether, and how, the findings of this study are likely to generalize to other species or experimental conditions, including any relevance to human biology (where appropriate).

2. Discussion of the data: At present, the discussion of the data is rather scant and should be expanded. Emerging evidence indicate a role of β2-adrenergic receptor in pain. More importantly, β2-adrenergic receptor agonist exerts potent analgesic effects (include but not limited to PMID: 19840789; 34673421; 27840124; 33829907 ), which seems controversial to this study.

3. Suitability of statistical tests: The statistical tests utilized are only suitable for use on normally distributed data, but the methods do not state whether normality tests were performed prior to subsequent analyses. Could the authors confirm that this was done and include a statement to this effect in the methods?

Author Response

Replies to the comments from the reviewer

Thank you very much for your time and his/her encouraging comments and constructive suggestions with regard to our manuscript. Those comments are helpful for us to revise and improve our manuscript. We have studied comments carefully and tried our best to revise and improve the manuscript according to the Reviewers’ comments. Revised portion is highlighted in RED FONT in the revised manuscript. We hope that the corrections will meet with approval. The main corrections in our manuscript and the responses to the Reviewers’ comments are as follows:

Reviewer 1:

Discussion

The design and reporting are generally good, but lack limitations and implications to conclude.

Response;

We agree with the reviewers’ concern and apologize for incomplete discussion. We have added some limitations and implications to Discussion Section.

Methods

  1. References are required to confirm that this is a systematic experimental method.

Response;

As the experimental method performed in this study was determined based on similar previous researches in this field [16, 18]. We added the description to Method Section (page 8-9).

  1. Please write the product name, company, and country for all devices, drugs, and software.

Response;

According to the suggestion by the reviewer, we added the description about the country of all devices, drugs, and software to Method Section (page 8-10).

Reviewer 2:

Underlying mechanisms of the protective effect of peripheral β2-adrenergic receptors blockers

Response;

We thank the reviewer for constructive comments and agree with the concern that more experiments are required to elucidate the detailed mechanism of the protective effect of the β2-adrenergic receptors blockade in AcGP model. This issue was added to the revised manuscript of Discussion (page 8, line 19-31).

  1. Reporting animal usage: Although this is generally well reported in this study, there are some extra details required in order to comply with ARRIVE guidelines (Percie du Sert, Hurst et al. 2020). To rectify this, the authors need to:

Response;

Thank you very much for your valuable comments. According your suggestions, we totally revised the manuscript, as follows;

  • Abstract:

State the species and strain of animals used in this study in the abstract

Response;

The species and strain of animals used (C57BL/6J mice) was added to Abstract Section (page 1).

  • Introduction:

Explain how and why the animal species and model being used can address the scientific objectives and the study’s relevance to human biology.

Response;

As suggested by the reviewer, we revised the Introduction of manuscript, (page2, lines 14-34).

  • Methods:

State the total number of animals used in the study, as well as clearly indicate group sizes for each experiment

Response;

Total number of mice and group sizes for each experiment were stated under 4.1. Animal housing and Table 1 (page 8-10).

the rationale for using the number and strain of animals chosen (e.g. sample size calculations).

Response;

Sample size and the reason why we chose this mouse strain followed the previous studies ([16], [44] and [45]). We added the new citation ([44] and [45]) to Reference Section and the corresponding description to Method Section (page 8).

randomisation

Response;

For the purpose of randomization, paw withdrawal thresholds of each animal were measured before experiments and animals were assigned so that the control and treatment groups had approximately equal mean paw withdrawal thresholds. We added the corresponding description to Method Section (page 8).

Blinding

Response;

For the blinding, investigators in charge of project and experiment designing and conduct of experiment are independent to each other. Data analyses after the experiments were performed by all authors in the presence of non-authors. We added the corresponding description to Method Section (page 8).

  • Results:

For each experimental group, report any animals, experimental units or data points not included in the analysis and explain why. For each experiment conducted, including independent replications, report: a. Summary/descriptive statistics for each experimental group, with a measure of variability where applicable (e.g. mean and SD, or median and range). b. If applicable, the effect size with a confidence interval.

Response;

The number of animals excluded in the analysis and the reason have been reported in table 1. The sample size of Figure 3C in legend have been revised. We added mean, SEM and p value to Result Section (page 3-5).

  • Discussion:

Comment on whether, and how, the findings of this study are likely to generalize to other species or experimental conditions, including any relevance to human biology (where appropriate). 

Response;

We have added the description about possibility of generalization to the first paragraph of Discussion Section (page 5).

  1. Discussion of the data: At present, the discussion of the data is rather scant and should be expanded. Emerging evidence indicate a role of β2-adrenergic receptor in pain. More importantly, β2-adrenergic receptor agonist exerts potent analgesic effects (include but not limited to PMID: 19840789; 34673421; 27840124; 33829907), which seems controversial to this study.

Response;

We are also interested in this difference. The possible major reason may be related the difference between neuropathic pain and AcGP, a fibromyalgia-like generalized pain disease. More details are discussed in the discussion (page 8, line 19-31).

  1. Suitability of statistical tests: The statistical tests utilized are only suitable for use on normally distributed data, but the methods do not state whether normality tests were performed prior to subsequent analyses. Could the authors confirm that this was done and include a statement to this effect in the methods?

Response;

In the present study we performed Shapiro-Wilk test to confirm normality of each data prior to ANOVA. We have added the description under the Statistical analysis of Method Section (page 10)

Reviewer 3 Report

The main question raised by the research is that administration of selective β2-blockers, including one with only peripheral action, inhibites the development but does not reverse the maintenance of pain-like behavior in saline-treated C57BL/6J mice.

The introduction is extensive and satisfactory for the subject.

The text has already been carefully corrected and I agree with the corrections. The way of presenting the work is unorthodox.

First the results are presented, then the material and method are presented. Conclusions from the study are summarized in an abstract and are not found in the main text. The discussion also precedes the reference to the sample and methods.

Regardless of whether one agrees with the methodology and the discussion, the standard structure of the text should be adhered to.

References should be corrected as indicated.

Author Response

Reviewer 3:

The main question raised by the research is that administration of selective β2-blockers, including one with only peripheral action, inhibites the development but does not reverse the maintenance of pain-like behavior in saline-treated C57BL/6J mice.

The introduction is extensive and satisfactory for the subject.

The text has already been carefully corrected and I agree with the corrections.

Response; We thank the reviewer for understanding our revised manuscript.

The way of presenting the work is unorthodox.

First the results are presented, then the material and method are presented. Conclusions from the study are summarized in an abstract and are not found in the main text. The discussion also precedes the reference to the sample and methods.

Regardless of whether one agrees with the methodology and the discussion, the standard structure of the text should be adhered to.

Response; We used the official template from International Journal of Molecular Sciences to prepare our manuscript. According to the instructions from the journal, Conclusions Section is not mandatory. Therefore, we wrote a sentence similar to the conclusions of the present study in the last paragraph of Discussion (page 8) and deleted Conclusions Section instead. Furthermore, according to the text structure instructed by the journal, we believe that there is no problem with the style that Material and Method Section comes after Result Section

References should be corrected as indicated.

Response; We deeply apologize for some careless mistakes in the style of some references (the references with minor style modifications are highlighted in BLUE FONT in page 12 of the revised manuscript). In addition, we noticed a mistake in the citation of No. 10 (original revised version at Round 1). Therefore, we replaced the original No. 10 citation (Abott. Nature 2010) with the new No. 14 citation (Sulka et al. Muscle Nerve 2001) at revised version of Round 2.

Reviewer 4 Report

This interesting and well written study has been focused on the role of splenocytes in eliciting fibromyalgia pain symptoms, and on the imbalance between the adrenergic and cholinergic signaling systems in this process; which is an interesting exploration avenue that had been previously developed by this research group and is now followed and strengthened, which is a strength point of this manuscript. However, the entire study is focused on proteins alone as reflecting these biological impacts, which is a weakness point since numerous small and long, coding and non-coding RNAs have been found to be functionally involved in creating and exacerbating the pain symptoms of fibromyalgia. Additionally, the topic of electrophysiology differences in peripheral cells of FMG patients is missing, which is a pity as it may be highly relevant to the findings presented here.

 To address these two issues, the authors may wish to add relevant text addressing the electrophysiology aspects of FMG and their potential relevance to the immune system issues covered in this study; additionally, the putative regulatory roles of RNAs in FMG should be covered both in the introduction and the discussion.

Author Response

Reviewer 4:

This interesting and well written study has been focused on the role of splenocytes in eliciting fibromyalgia pain symptoms, and on the imbalance between the adrenergic and cholinergic signaling systems in this process; which is an interesting exploration avenue that had been previously developed by this research group and is now followed and strengthened, which is a strength point of this manuscript.

Response; We thank the reviewer for understanding our revised manuscript.

However, the entire study is focused on proteins alone as reflecting these biological impacts, which is a weakness point since numerous small and long, coding and non-coding RNAs have been found to be functionally involved in creating and exacerbating the pain symptoms of fibromyalgia.

Response; We agree with the reviewer’s concern that our study seems lack the description regarding to RNAs biology in FM pathology. We found many reports about RNAs relevant to FM pathology. Among them, Qiu et al. Clin Exp Rheumatol. 2021 and Dolcino et al. J Clin Med. 2020 reported that differentially expressed genes, miRNAs and long non-coding RNAs associated with dysregulated immune system were found between FM patients and healthy control. This may play an important role in the pathogenesis of FM. However, it remains elusive whether their expressions also change in AcGP model and relate to splenocytes modulated by sympathetic input.

We have referred the above papers (reference No. 12, 13) and added a sentence (Recent findings indicated… …pathogenesis of FM) in Introduction (line 57-60 in page 2). Moreover, we have written about the necessity of future investigations to clarify transcriptional changes in splenocytes in AcGP model in Discussion (line 238 in page 8).

Additionally, the topic of electrophysiology differences in peripheral cells of FMG patients is missing, which is a pity as it may be highly relevant to the findings presented here.

Response; We thank the reviewer for pointing out the important aspects. We found some electrophysiological studies of FM (Serra et al. Ann Neurol. 2014, Teng et al. J Formos Med Assoc. 2021). They show spontaneous activity in silent C nociceptors and increased sensory nerve excitability in patients with FM, suggesting that peripheral sensitization is one of the pathological factors in FM.

We have referred these papers (reference No. 7 and 8) and added the description (On the other… …of FM patients) in Introduction (line 49-51 in page 2).

Also, as the reviewer pointed out, an enhancement of peripheral sensory neuronal activity is a possible mechanism which explain how splenocytes of AcGP mice contribute to pain development.

We have referred a study about peripheral sensitization in AcGP model (reference No. 44) as well as studies about central sensitization (reference No. 42 and 43), and the possible immune effects on them in Discussion (line 251-254 in page 8).

To address these two issues, the authors may wish to add relevant text addressing the electrophysiology aspects of FMG and their potential relevance to the immune system issues covered in this study; additionally, the putative regulatory roles of RNAs in FMG should be covered both in the introduction and the discussion.

Response; As above, we have added some references about the electrophysiological aspects and the putative regulatory roles of RNAs in FM and discussed their potential relevance to the dysregulated immune cells in AcGP model.

Round 2

Reviewer 2 Report

The results are preliminary. The underlying mechanisms are warranted.

Author Response

Reviewer 2:

The results are preliminary. The underlying mechanisms are warranted.

Response; We thank the reviewer for their understanding of the underlying mechanisms we have elucidated. We will continue to work hard to further elucidate the pathophysiology.

Reviewer 3 Report

The authors accepted corrections other than those characterizing the structure of the paper. I believe that the separation of method and statistical analysis from conclusions should be clear. The discussion and conclusions section should be separate and follow the others.

Author Response

Reviewer 3:

The authors accepted corrections other than those characterizing the structure of the paper. I believe that the separation of method and statistical analysis from conclusions should be clear. The discussion and conclusions section should be separate and follow the others.

Response; We are sorry for insufficient revisions in the last submitted manuscript. We agree with the reviewer’s concern and separated conclusions from Discussion section. Namely we deleted some sentences corresponding to the conclusions of this study in the last paragraph of Discussion (page 8) and newly made Conclusions Section (page 11).
